# Development of Thermosensitive and Mucoadhesive Hydrogel for Buccal Delivery of (*S*)-Ketamine

**DOI:** 10.3390/pharmaceutics14102039

**Published:** 2022-09-24

**Authors:** Agathe Thouvenin, Balthazar Toussaint, Jelena Marinovic, Anne-Laure Gilles, Amélie Dufaÿ Wojcicki, Vincent Boudy

**Affiliations:** 1CNRS, Inserm, UTCBS, Université Paris Cité, F-75006 Paris, France; 2Département Recherche et Développement Pharmaceutique, Agence Générale des Équipements et Produits de Santé (AGEPS), AP-HP, F-75005 Paris, France

**Keywords:** poloxamer, alginate, mucoadhesion, drug release

## Abstract

(*S*)-ketamine presents potential for the management of acute pain and, more specifically, for the prevention of pain associated with care. However, the administration route can be a source of pain and distress. In this context, a smart formulation of (*S*)-ketamine was designed for buccal administration. The combination of poloxamer 407 and sodium alginate enables increased contact with mucosa components (mucins) to improve the absorption of (*S*)-ketamine. In this study, rheological studies allowed us to define the concentration of P407 to obtain a gelling temperature around 32 °C. Mucoadhesion tests by the synergism method were carried out to determine the most suitable alginate among three grades and its quantity to optimize its mucoadhesive properties. Protanal LF 10/60 was found to be the most effective in achieving interaction with mucins in simulated saliva fluid. P407 and alginate concentrations were set to 16% and 0.1%. Then, the impact of P407 batches was also studied and significant batch-to-batch variability in rheological properties was observed. However, in vitro drug release studies demonstrated that this variability has no significant impact on the drug release profile. This optimized formulation has fast release, which provides potential clinical interest, particularly in emergencies.

## 1. Introduction

Common medical procedures used to diagnose and treat patients can cause pain and anxiety, especially in children. Examples include intravenous access, laceration repairs, and orthopedic procedures. This acute pain related to care can have long-term negative consequences for children, such as phobic behavior towards care and caregivers. Different methods and products are recommended for the prevention and relief of pain associated with care in children [1,2]. Nitrous oxide is the reference product to prevent pain related to painful procedures and care. It has analgesic, anxiolytic, and euphoric effects and an excellent safety profile. However, as a sole agent, it does not reliably produce adequate procedural conditions and is associated in many cases with an opioid or locoregional anesthesia [3]. To manage the cases in which nitrous oxide cannot be used or is not efficient, ketamine has presented potential for the management of acute pain at low doses but it is intravenously (IV) or intramuscularly (IM) injected [4,5]. However, these routes of administration can be a source of pain and are therefore not optimal for the prevention of pain associated with care.

Ketamine is a non-competitive antagonist of post-synaptic NMDA (N-Methyl-D-Aspartate) receptors [6]. This active substance is a general anesthetic in high doses and an analgesic in low doses [7]. It shows a short onset of action from 1 to 2 min after an IV injection [8]. Its use since the 1980s has provided extensive pharmacological and toxicity data. For an analgesic effect, the recommended dose of ketamine IV is 0.5 to 1.0 mg/kg [7] and these analgesic doses are ten times lower than the anesthetic doses commonly used. Ketamine is a chiral molecule and the (*S*)-enantiomer is a four-times more potent anesthetic and analgesic [9,10]. The IV route is widely used because the oral bioavailability is low (8–11% [11,12]) due to the hepatic first-pass effect, but recent studies showed that ketamine can easily pass barriers such as nasal mucosa [13].

Today, it would be interesting to expand the therapeutic options with a new drug formulation allowing the fast acting of the drug, a sufficient analgesic level with limited side effects, and a suitable medication for the pediatric population (i.e., the control of the dose by modification of administered volume according to the age/weight/corporal surface, easy administration, without pain related to administration or tissue infraction). Thus, we are interested in the formulation of (*S*)-ketamine for buccal administration (i.e., transmucosal route, thus avoiding the first-pass metabolism). For buccal delivery, mucoadhesive forms such as tablets, patches, or films were developed to increase the bioavailability of the active substances by reducing salivary losses and by optimizing contact and thus absorption. The disadvantage of these galenic forms is the difficulty of adapting the dose administered to the patient’s weight. Liquid forms are more frequently adopted, especially in children.

A thermogelling solution using poloxamer 407 (P407) was designed. P407 is a synthetic copolymer composed of ethylene oxide (EO) and propylene oxide (PO) monomers arranged in a triblock structure: EO_x_-PO_y_-EO_x_. The thermogelification phenomenon is perfectly thermoreversible and is characterized by a gelling temperature (T_sol-gel_). Below this temperature, the sample remains liquid, and, above it, the solution becomes semi-solid. Its liquid form allows the easy adjustment of the administered dose and the homogeneous distribution of the active molecule in the hydrogel. In its gelling form, P407-based hydrogel allows a prolongation of the residence time of the formulation at the administration site. Gelling form can be used to prolong the drug release (decrease in diffusion rate or slow erosion) or to enhance the absorption of the active substance in the case of local administration. This is why the addition of a mucoadhesive agent in the formulation is essential to promote the persistence of the form within the buccal cavity. The association of this poloxamer with a bioadhesive agent is of interest for transmucosal routes such as vaginal, rectal, or buccal routes in order to increase the residence time of the product in the mucosa [14]. Alginate is a naturally linear polysaccharide. This polymer consists of 1,4 β-D-mannuronic acid (M) and 1,4 α-L-guluronic acid (G) residues. There are different grades of alginate based on their composition of M and G residues. All residues in alginate interact with multivalent cations to form a gel, especially calcium, but the strength of the interaction increases with the proportion of G residues [15]. Alginate has also the ability to interact with mucins present on the mucosal surface [16].

This paper describes the design of a novel thermosensitive hydrogel for the buccal delivery of (*S*)-ketamine with an optimized gelling temperature, mucoadhesive properties, and immediate drug release. In this paper, we focused on the determination of the P407 concentration to obtain the targeted gelling temperature, the determination of the interaction between different grades of alginate and mucins, the (*S*)-ketamine release study from the optimized formulation, and the impact of the batch-to-batch variability of P407 on formulation properties.

## 2. Materials and Methods

### 2.1. Chemicals

(*S*)-ketamine hydrochloride (SK) was purchased from Seqens (Lahr/Schwarzwald, BW, Germany). Three batches of poloxamer 407 (P407, Kolliphor 407) were obtained from BASF (Geismar, LA, USA). “Low-viscosity” alginates were used (viscosity < 100 mPa·s at 20 °C for a 1% alginate solution). Three grades of sodium alginate, based on the proportions of 1,4 β-D-mannuronic acid (M) and 1,4 α-L-guluronic acid (G) residues, were bought from FMC Biopolymer (Cork, Ireland): Protanal LF 10/60 (PRO), Manucol DH (MAN), and Keltone LVCR (KEL). Water used here was sterile water, Versylène^®^, obtained from Fresenius Kabi (Sevres, France). Sodium chloride (NaCl), potassium dihydrogen phosphate (KH_2_PO_4_), and sodium bicarbonate (NaHCO_3_) were purchased from Cooper (Melun, France). Potassium chloride (KCl) and di-potassium hydrogen phosphate (K_2_HPO_4_) were obtained from Kirsch Pharma (Salzgitter, Germany). Calcium chloride (CaCl_2_, 2H_2_O) was purchased from VWR International (Fontenay-sous-Bois, France). The porcine gastric mucin type II was purchased from Sigma-Aldrich (St. Louis, MO, USA).

### 2.2. Preparation of Hydrogels

Concentrations of each component were defined by the mass of the component over the total mass of the hydrogel (weight/weight ratio, *w/w*). Only percentages of P407 were defined by the mass of P407 relative to the mass of water contained in the hydrogel (w_P407_/w_water_). Hydrogels contained SK (9.23% *w/w*), P407 (from 15.0 to 17.0% w_P407_/w_water_), and sodium alginate (from 0.10 to 0.20% *w/w*). The concentration of SK was set at 9.23% *w/w* to obtain a dose of 13 mg (0.5 mg/kg), for an administrated volume of 140 µL corresponding to one spray. Three different batches of the same grade of P407 (L1, L2, L3) from the same supplier, and three grades of alginate (PRO, MAN, KEL), were studied. The properties of the three grades of alginate are presented in Table 1. All conditions are listed in Table 2.

SK was rapidly dissolved in water before adding alginate under magnetic stirring at 350 rpm for approximatively 15 min at room temperature. P407 was dissolved by the cold method [20] directly in the solution containing SK and alginate at 4 °C for 24 h to ensure complete dissolution. Each solution was stored at 4 °C until rapid use and was placed under magnetic stirring at room temperature for 30 min before tests.

### 2.3. Preparation of Artificial Media for Hydrogel Studies

#### 2.3.1. Artificial Saliva Preparation

To reproduce the pH conditions and ionic composition of saliva, an artificial saliva solution composed of 5 mM KH_2_PO_4_, 15 mM KCl, 1 mM CaCl_2_, and 5 mM NaHCO_3_ was prepared (pH 6.8) [21]. Briefly, for 1 L of artificial saliva, 0.680 g of KH_2_PO_4_, 1.130 g of KCl, 0.150 g of CaCl_2_, and 0.420 g of NaHCO_3_ were dissolved at room temperature under magnetic stirring at 300 rpm.

#### 2.3.2. Mucin Solution Preparation

Mucin was dispersed slowly under magnetic stirring (500 rpm) in phosphate buffer (0.1 M, at pH 6.8) or artificial saliva previously described to obtain 10% *w/w* stock solutions. Dilutions were made in phosphate buffer or artificial saliva to obtain mucin solutions at 2, 3, 4, and 5% *w/w* in testing samples. Stock solutions and mucin solutions were stored at 4 °C until rapid use.

### 2.4. Gelling Temperature Study

Rheological studies were carried out on an Anton Paar MCR102 Rheometer (Graz, Austria) with a cone–plate geometry (diameter: 50 mm, cone angle: 1°, gap: 0.1 mm), providing a homogeneous shear of the samples. All data were analyzed using the Anton Paar RheoCompass™ software version 1.25 (Graz, Austria) associated with the rheometer. The gelling temperature (T_sol-gel_) of the samples was determined by oscillatory measurements. An amplitude sweep and a frequency sweep were applied to determine the linear viscoelastic region. All subsequent measurements of the storage modulus (G′) and loss modulus (G″) were run within the linear viscoelastic region at amplitude of 0.1% and a frequency of 1 Hz, where G′ and G″ remained invariant and the sample did not undergo structural modifications. All measurements were performed using a temperature sweep analysis over temperatures ranging from 20 to 40 °C and a heating rate of 1 °C/min. The gelling temperature was defined as the cross-over point where G′ and G″ moduli are equal (G′ = G″).

### 2.5. In Vitro Evaluation of Mucoadhesion

The mucoadhesive properties of alginate were evaluated by rheological synergism methods. The rheological measurements were carried out on an Anton Paar MCR102 Rheometer (Graz, Austria) with plate–plate geometry (diameter: 50 mm, gap: 0.5 mm). All measurements were performed at a constant temperature of 37 ± 0.05 °C.

#### 2.5.1. Flow Measurements

Flow measurements was applied to evaluate mucin–polymer interactions: this method is based on the evaluation of the rheological synergism existing between the mucoadhesive polymer and mucin [22]. Two solutions were used to identify the impact of calcium on the mucoadhesive action of alginate: a phosphate buffer (0.1 M) at pH 6.8 and the artificial saliva solution. Sodium alginate (PRO, 0.10% *w/w*) was studied alone in artificial saliva or phosphate buffer. Different concentrations of mucin (from 2 to 5% *w/w*) dispersed in artificial saliva or phosphate buffer were tested in order to assess the impact of mucin amount on the mucoadhesion mechanism. A shear rate ranging from 0.1 to 100 s^−1^ was applied and the viscosity of each solution was determined.

The viscosity of alginate, mucin, and alginate–mucin mixture solutions was measured to determine the interaction parameter (η*_b_*), which was calculated using the following equation:(1)$$\eta_{b}\;=\eta_{t}-\left(\eta_{p}+\eta_{m}\right)$$
where η*_t_* represents the viscosity of the solution containing alginate and mucin; η*_p_* and η*_m_* are the respective viscosities of the alginate and mucin.

#### 2.5.2. Oscillatory Measurements

Three grades of alginate (PRO, KEL, and MAN) were studied to evaluate the mucoadhesion properties as a function of grade. The polymer solutions containing various concentrations of alginate (0.10, 0.15, and 0.20% *w/w*) and a fixed percentage of P407 (16.0% w_P407_/w_water_) were tested (Table 1).

Mucoadhesive properties of alginate were evaluated by the rheological synergism method described by Bassi da Silva et al. [23]. Oscillatory frequency sweep measurements (0.1–10 Hz) were performed within the linear viscoelasticity range. Storage (G′) and loss (G″) moduli and loss factor (tan δ = G″/G′) were measured for all samples after an equilibration time of 60 s to ensure temperature adaptation. Three solutions were tested to determine the rheological synergism parameter: mucin solution, polymer solution, and polymer–mucin mixture. The mixture was obtained by diluting polymer solution with mucin solution in the ratio of 5:1 (*v/v*). Artificial saliva was used to maintain the same dilution of all samples. The rheological synergism parameter (ΔG′) is defined by the following equation:(2)$$\Delta G^{\prime }\;={\;G^{\prime }}_{t}\;-\;\left({G^{\prime }}_{m}+{\;G^{\prime }}_{p}\right)$$
where G′*_t_* is the elastic modulus of the mixture and G′*_p_* and G′*_m_* represent the elastic moduli of polymer and mucin, respectively. However, the small elastic modulus of mucin can be considered negligible when compared to the elastic modulus of polymer [24]. Equation (2) was simplified, and rheological synergism parameters were calculated using the following equation:(3)$$\Delta G^{\prime }\;=\;{G^{\prime }}_{t}\;-\;{G^{\prime }}_{p}$$
where G′*_t_* is the elastic modulus of the mixture and G′*_p_* is the elastic modulus of polymer. The values obtained at the intermediate value of 1.0 Hz have been chosen to compare results.

### 2.6. In Vitro Drug Release

SK release from SK/PRO/P407-L1, -L2, and -L3 hydrogel was evaluated using USP-4 apparatus Sotax CE7 Smart (Sotax AG, Nordring, Switzerland) equipped with seven standard cells with a diameter of 22.6 mm. SK was monitored with an UV–VIS spectrophotometer, Lambda 25 (Perkin Elmer, Waltham, MA, USA), directly connected to the USP-4 apparatus, allowing direct on-line analysis. In each cell, a ruby bead of 5 mm in diameter and glass beads of 1 mm in diameter were placed in the apex of the flow-through cell to ensure laminar flow. A sample of 420 μL hydrogel was placed into the glass bead bed. Before starting the test, cells were placed in a 37 °C oven until the solution gelled. To respect the sink condition, 75 mL artificial saliva was used for each cell to reach a maximal SK concentration of 0.52 mg/mL. This volume was pumped through each cell with a flow rate of 8 mL/min. Temperature
of 37.0 ± 0.5 °C was maintained throughout the study. The concentration of SK was determined at regular intervals (every 2 min, for 120 min) at 269 nm, corresponding to the wavelength of maximal absorption. Drug release profiles were obtained by cumulative percentage of SK release (Mt/M∞, %) versus time (t, min).

Dissolution profiles were analyzed with the DDSolver software [25]. The similarity factor (ƒ_2_), as described in Equation (4), was used to compare drug release profiles [26]:(4)$$f_{2}=50\times \;\log\;\left\{\;{\left[\;1+\frac{1}{N}\left(\sum_{t=1}^{N}{\left(R_{t}\;-\;T_{t}\right)}^{2}\;\right)\right]}^{-0.5}\times 100\;\right\}$$
where *R_t_* is the amount of released drug on the reference formulation, *T_t_* is the amount of released drug on the test formulation, and *N* is the number of experimental data values. The profiles were considered similar if ƒ_2_ was between 50 and 100.

Different mathematical models were used on dissolution profiles to investigate dissolution phenomena. Equations and parameters of the models are detailed in Table 3. The drug release kinetics were analyzed by fitting the experimental data to the kinetic models. The values of kinetic parameters were calculated, and adjusted coefficients of determination (R^2^_adj_) and Akaike information criterion (AIC) were determined for each model.

When applying the Peppas–Sahlin model, the percentages of diffusion (Equation (10)) and the percentage of erosion (Equation (11)) at time t were calculated to quantify the contribution of the two mechanisms [27]:(10)$$\frac{Mt_{diffusion}}{M\infty }\;\left(\%\right)=\frac{1}{1+\left(k_{2}/k_{1}\right)t^{1/2}}\times 100$$
(11)$$\frac{Mt_{erosion}}{M\infty }\;\left(\%\right)=\left(1-\frac{1}{1+\left(k_{2}/k_{1}\right)t^{1/2}}\;\right)\times 100$$

### 2.7. Statistical Analysis

Statistics were performed on the results observed on three independent samples. Two-way ANOVA and Bonferroni’s multiple comparisons tests were conducted using the Prism 7.00 software (GraphPad, Northside, CA, USA). A significant difference was accepted when the significance level was less than 0.05 (*p*-value < 0.05). Levels were *: *p* < 0.05; **: *p* < 0.01; ***: *p* < 0.001; ****: *p* < 0.0001.

## 3. Results and Discussion

### 3.1. Relation between Concentration of Poloxamer and Gelling Temperature

The concentration of P407 is determined according to the targeted gelling temperature. In this context, the targeted gelling temperature was defined between 30 and 35 °C to obtain a liquid form at room temperature and a gelled form on contact with buccal mucosa.

The relation between gelling temperature and P407 concentration was investigated by adding concentrations of P407 from 15.0 to 17.0% (w_P407_/w_water_) in SK and PRO solution (SK/PRO/P407_1–5_ hydrogels). A linear correlation between gelling temperature and P407 concentration was observed (R^2^ = 0.994) (Figure 1a). The gelling temperature increased with the diminution of the P407 concentration. A solution of P407 at 15.0% w_P407_/w_water_ in the presence of SK and PRO presented a gelling temperature of 34.2 ± 0.4 °C, whereas a gelling temperature of 27.8 ± 0.9 °C was measured for the same concentration of P407 alone. The high concentration of SK induced a modification of the T_sol-gel_ because its addition is unfavorable to the interactions between the polymer chains, thus delaying the ability of the formulation to form a gel. On the contrary, the addition of alginate causes a decrease in T_sol-gel_ by promoting the formation of P407 micelles. The elastic component is weaker due to a decrease in the interactions between the P407 micelles caused by the presence of the alginate chains [28]. The association of the three components has thus allowed us to find a balance by obtaining suitable rheological properties in terms of T_sol-gel_ (30 to 34 °C), storage modulus G′ (around 5000 Pa), and viscosity in solution state (lower than 200 mPa·s at 20 °C). The difference between the T_sol-gel_ and the temperature of the oral mucosa ensures a phase transition of the hydrogel at the time of administration and guarantees the administration of the product in solution form under storage conditions at a temperature below 30 °C. Based on this result, the P407 concentration to obtain a gelling temperature of 31.0 °C was determined at 16.0% w_P407_/w_water_ (Figure 1a). To confirm this result, three samples were prepared with the same batch of P407 (L1). This experiment demonstrated a satisfactory gelling temperature of 30.8 ± 0.2 °C (Figure 1b).

### 3.2. Mucoadhesive Properties

Sodium alginate was chosen as the mucoadhesive agent in this formulation. Sodium alginate exhibits interesting properties and compatibility with P407, as discussed in the review written by E. Giuliano et al. [29]. The addition of alginate maintains the low viscosity of the solution at room temperature in the absence of calcium, which guarantees suitable properties for the administration of the product. In situ, alginate can provide mucoadhesive properties to the formulation because it interacts with mucins on the surfaces of mucous membranes. These properties have been demonstrated for oral, nasal, or ophthalmic forms for concentrations often higher than 1% *w/w* [30,31]. The addition of a similar concentration would have led to the loss of the thermogelling properties of the solution by the disorganization of the P407–micelle interaction. The lowest alginate concentration (0.10% *w/w*) was studied first for mucoadhesion evaluation because this level influences the rheological properties of the hydrogel to a lesser degree. The mucoadhesive properties of different grades of alginate were then studied at this concentration using a rheological synergism method.

Three types of alginate (PRO, MAN, KEL) were studied because they present different guluronate (G) monomer content. The G monomer content is often defined by the mannuronate/guluronate ratio (M/G ratio) and is considered high if the M/G ratio is below 0.9. Only PRO has a high G residue level (Table 3). The rheological synergism method was used to compare their mucoadhesive properties.

#### 3.2.1. Mucoadhesive Properties of Alginate

The viscosity of the alginate solution (PRO) at 0.10% *w/w* was evaluated in phosphate buffer and in artificial saliva, which thus contains calcium. Using these two buffered media, the change in viscosity of the alginate solution in the presence of calcium ions in saliva can be identified. Alginate is capable of complexing calcium ions Ca^2+^, forming an “egg box” structure [15]. This conformation of the polymer chains leads to an increase in viscosity. The viscosities of alginate solutions in phosphate buffer or in artificial saliva were not significantly different (1.5 ± 0.9 mPa·s and 1.9 ± 0.2 mPa·s at 1 s^−1^, respectively) and are close to the viscosity of an aqueous solution. The presence of calcium in artificial saliva did not modify significantly the viscosity of the alginate solution in our conditions (1 mM of calcium in artificial saliva, alginate (PRO) at 0.10%).

Afterwards, the viscosities of the alginate solution alone, the mucin solutions, and the alginate–mucin mixtures were measured. The values of viscosities measured at a shear rate of 1 s^−1^ are shown in Table 4.

The viscosities of mucin solutions depend on the concentration of mucin and the aqueous media (*p* < 0.001 and *p* = 0.005, respectively). The contribution of ions by the artificial saliva, such as monovalent or bivalent cations, seems to be at the origin of the viscosity increase. Mucins are long, negatively charged chains, which gives them an extended conformation. Mucins can form a gel when their concentration is higher than 20 mg/mL by entanglement of the chains [32]. The presence of cations allows for the negative charges to be shielded, which leads to an electrostatic repulsion decrease and favors hydrophobic interactions between the mucin chains [33]. These weak interactions can explain the viscosity decrease when the shear rate increases (Figure 2).

The interaction parameter was calculated according to Equation (1) as a function of mucin concentration (Table 4). The interaction parameter between mucin and alginate is largely higher in the artificial saliva medium. The addition of divalent cations seems to promote the interaction between mucin and this polymer. Fuongfuchat et al. observed similar results with 3 mM Ca^2+^ added to different mixed solutions of mucin and alginate [34].

The interaction parameter was also calculated from the viscosities measured at shear rates of 10 s^−1^ and 100 s^−1^ (Figure 2). The interaction parameter decreased with increasing shear rates. This decrease is more significant with increasing mucin concentrations. This study demonstrated the presence of an interaction between alginate and mucin. This interaction is dependent on the presence of divalent cations in the medium used and the shear rate when the alginate proportion remains constant (PRO at 0.10%). Based on these results, the mucoadhesive agent was then studied in artificial saliva with a concentration of mucin at 5% (*w/w*) to promote the interaction and to enable the comparison between different formulations. Viscosity measurements showed a decrease in the interaction parameter as a function of the shear rate. Flow measurements are destructive, especially for weak bonds, and can undervalue the interaction parameter. Therefore, the choice of the mucoadhesive agent was made from oscillation measurements.

#### 3.2.2. Mucoadhesive Properties of P407–Alginate Association

A rheological method was carried out to determine which grade of alginate confers the greatest mucoadhesive property to the formulation. The storage moduli were measured for P407, P407/PRO, P407/KEL, and P407/MAN solutions according to different percentages of alginate (from 0.10 to 0.20%), with or without mucin at 5%. These initial results showed two different G′ modulus curve profiles (Figure 3a). P407/PRO, P407/KEL, and P407/MAN presented a similar G′ modulus curve profile, while the addition of alginate in a P407 suspension leads to a decrease in the G′ modulus in a homogeneous way, independently of the oscillation frequency. In the presence of mucin, the G′ modulus decreases, especially when the oscillation frequency is low.

The interactions between mucins and mucoadhesive polymers are related to different mechanisms. Oscillation analysis differentiates physical entanglements from weak interactions such as hydrogen bonds [23,35]. The decrease in storage modulus at low oscillation frequencies and the increase in loss factor (Figure 3b) may be due to interpenetration between the P407 polymer chains and the peptide chains of the mucin protein, leading to disorganization of the P407 micelles. This phenomenon disappears when the frequency increases and the storage modulus G′ returns to a value close to that of the solutions without mucin. In the presence of alginate, the storage modulus G′ is constantly decreased as a function of the oscillation frequency compared to the P407 suspension. This indicates a possible interaction of alginate with P407 micelles. This phenomenon was also observed during rheological tests, since the addition of alginate leads to a decrease in the T_sol-gel_ of the hydrogel. Furthermore, the presence of alginate leads to a more important decrease in the storage modulus G′, which can be reflected by an interaction between alginate and mucin.

To compare the three grades of alginate, ΔG′ (the interaction parameter) was calculated with Equation (3) for each solution, and the results are presented in Figure 4 (detailed values of G′*_p_*, G′*_t_*, and ΔG′ are reported in Appendix A).

The P407–mucin mixture showed a negative interaction parameter that could be the result of an interaction between the polymer and mucin. This interaction results in reduced organization of the P407 micelles and decreased solid behavior of the hydrogel. The P407/PRO_10–20_ solutions showed an evolution of the interaction parameter as a function of the alginate concentration. This reduction in the interaction parameter was proportional to the concentration of PRO. However, P407/KEL_10–20_ and P407/MAN_10–20_ solutions showed no significant difference in ΔG′. The main difference between the three grades of alginate is the level of G residues present in their chemical structures. PRO has a high ratio of G residues within its chemical structure, while KEL and MAN have a majority of M residues. These differences in M/G ratio may explain the observed differences in ΔG′ [16]. Thus, the potential of PRO as a mucoadhesive agent was confirmed at low concentrations (0.10 to 0.20%, *w/w*).

### 3.3. Effect of P407 Batches on Rheological Parameters

The variability of the rheological properties of hydrogels when using different batches of P407 has been assessed in order to verify the reproducibility of the formulation attributes. Thereby, gelling temperature and viscoelastic properties (storage and loss moduli G′ and G″, respectively, loss factor) were studied on two other batches of P407 (L2 and L3) to determine the batch-to-batch variability (Figure 5). SK/P407/PRO_10_ hydrogels containing L2 and L3 P407 batches showed a gelling temperature above 30.1 ± 0.6 °C and 33.1 ± 0.2 °C, respectively. L1 and L2 presented similar storage moduli G′ above 5700 ± 300 Pa and 6900 ± 800 Pa, respectively. The loss factor (tan δ), representing the ratio between loss and storage moduli, was less than 1 for the three batches of P407 used. It indicated the predominance of the storage modulus for L1 and L2 (0.40 ± 0.03 and 0.42 ± 0.02, respectively). However, L3 presented different rheological characteristics. Indeed, the storage modulus G′ of L3 decreased significantly (1400 ± 200 Pa). The loss factor was then higher and approached 1 (0.95 ± 0.03). These results obtained with the three batches of P407 showed relative standard deviations of 54% for elastic modulus, 31% for viscous modulus, and 46% for loss factor. This study highlighted a significant difference in rheological properties between the three batches of P407, particularly for gelling temperature (*p* < 0.001), when using the previously selected concentration of 16% w_P407_/w_water_ (Section 3.1). This concentration ensures a gelling temperature in an acceptable interval between 30 and 35 °C according to storage and administration conditions. However, the important difference in storage modulus could affect the dissolution characteristics of the hydrogel. Thus, the batch-to-batch variability was also studied using a dissolution test to evaluate the impact on drug release, as discussed in Section 3.4 below.

### 3.4. Effect of P407 Batches on Drug Release Profiles

In vitro tests were carried out to study the phenomena of SK release. The dissolution test allowed us to highlight the mechanisms at the origin of drug release: diffusion and/or erosion. These two mechanisms have been described in the literature and differ according to the quantity of P407 and the rheological characteristics of the hydrogel [27,36].

Dissolution tests were performed on the SK/P407/PRO hydrogel prepared with three different batches of P407 (L1, L2, and L3) to evaluate the impact of inter-batch rheological differences on SK release kinetics. A flow-through USP-4 apparatus was used and the SK released fraction was monitored (Figure 6). Spectral interferences were first investigated and no spectral interference was observed at 269 nm between SK and all other components at varying proportions (Appendix A). These results showed 30% release of SK in less than 10 min for the three hydrogels studied (Figure 6). SK/P407/PRO-L1 and -L2 showed similar release kinetics (80% of SK released in 32 ± 7 min and 30 ± 8 min, respectively) while a faster release was observed with the SK/P407/PRO-L3 hydrogel (80% of release in 23 ± 4 min).

To compare the profiles of each hydrogel, ƒ_2_ was calculated. If ƒ_2_ is between 50 and 100, the release profiles are similar. For the comparison of L1 and L2 batches, ƒ_2_ was close to 94.9 and showed high similarity between the two release profiles. This factor was lower for the comparison of L1 and L2 with L3 (56.5 and 58.3, respectively) but still indicated similarity with the other batches.

For each hydrogel, the release kinetics were modeled. The results of the model parameters, adjusted coefficient of determination (R^2^_adj_) and Akaike information criterion (AIC), are summarized in Table 5. Higuchi’s model does not appear to be suitable for modeling the release kinetics of SK based on R^2^_adj_ values. Thus, the release kinetics do not appear to be only related to the diffusion phenomenon [37]. The other models presented a relatively high goodness of fit, with a coefficient of determination close to 1 and a low AIC. The Makoid–Banakar model exhibited the best R^2^_adj_ for SK/P407/PRO-L1 and -L2 release profiles (1.000 and 0.999, respectively) and the lowest AIC (22.1 and 28.1, respectively). However, the best-fitting model for SK/P407/PRO-L3 release kinetics is the Korsmeyer–Peppas model. As the k parameter of the Makoid–Banakar model was almost zero, the equation became similar to the Korsmeyer–Peppas equation (Table 2). For these two models, it was observed that the values of n were comprised between 0.5 and 1 [38,39]. This observation indicated the superposition of diffusion and erosion phenomena (Figure 7). When applying the Peppas–Sahlin model, the diffusion phenomenon (k_1_) and the erosion phenomenon (k_2_) are dissociated. Here, the diffusion constant (k_1_) is higher than the erosion constant (k_2_), so the diffusion phenomenon was the major mechanism leading to the release of SK [40]. Figure 8 represents the contributions of the diffusion and erosion of formulations, and these contributions (in percentage) were calculated from Equations (10) and (11). The release mechanisms were similar for all formulations, with a predominant contribution of diffusion. However, a crossover in the diffusion and erosion contributions was observed for SK/P407/PRO-L1. These results are reliable with the values of the parameter k_1_, which is higher for SK/P407/PRO-L3 than for SK/P407/PRO-L1 or -L2.

All these results can be related to the rheological parameters of the P407 batches presented in Section 3.3. Indeed, the SK/P407/PRO-L3 formulation presented the lowest storage modulus, which may explain the increase in diffusion contribution and the decrease in erosion contribution. However, (*S*)-ketamine is a small molecule (molecular weight of 237.7 g/mol) and is freely soluble in water (solubility of ketamine hydrochloride is near 200 mg/mL). These physicochemical properties promote the diffusion of SK in the dissolution media. Thus, the difference in rheological properties had no significant impact on the release kinetics of SK.

## 4. Conclusions

A thermoresponsive hydrogel based on P407 was designed to develop a new dosage form of (*S*)-ketamine suitable for buccal administration. The association of (*S*)-ketamine (9.23% *w/w*), P407 (16% w/w_water_), and alginate (0.1% *w/w*) was studied to ensure an optimal gelling temperature around 31 °C. This formulation remains in liquid form at room temperature, allowing easy administration, and is in gel form on contact with buccal mucosa to extend the residence time. A mucoadhesive agent was incorporated into the P407-based hydrogel to optimize the residence time of this formulation on the buccal mucosa and then increase the absorption of SK through mucosa. The inclusion of alginate in this formulation increased the interaction with mucin and so suggests better adhesion to mucosa. Among the three grades of alginate, Protanal LF 10/60 showed different behavior in the interaction with mucin. This difference could be explained by the high ratio of the guluronate component in the alginate structure. Then, significant batch-to-batch variability of P407 in the rheological properties was observed. However, in vitro drug release studies demonstrated that this variability had no significant impact on the drug release profile (ƒ_2_ > 50). Mathematical models were also applied to understand the mechanisms at the origin of drug release. Two phenomena, diffusion and erosion, occurred in the hydrogel. Diffusion was the predominant mechanism related to the (*S*)-ketamine properties. For this type of molecule (i.e., small molecular weight, freely soluble in water), an adaptation of the P407 concentration as a function of batch would not be considered to compensate for the different rheological properties. This new formulation of (*S*)-ketamine is a promising alternative to address acute pain. The transmucosal route is interesting (1) to increase the bioavailability of a drug sensitive to first-pass metabolism, (2) to increase the onset of action when compared to per os administration, and (3) to avoid tissue infraction caused by IV or IM injections and thus superimposed pain and distress.

## Figures and Tables

**Figure 1 pharmaceutics-14-02039-f001:**
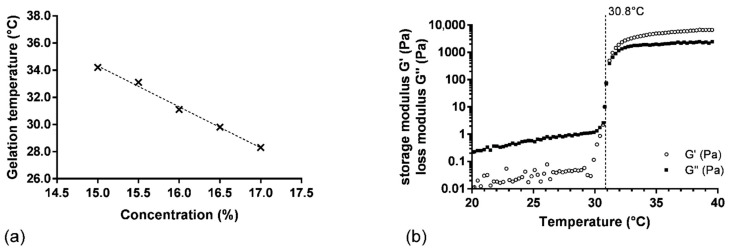
(**a**) Gelation temperature as function of P407 concentration (SK/PRO/P407_1–5_ hydrogels), the dash line represents the linear regression on experimental values (**×**); (**b**) variation in storage (G′) and loss (G″) moduli as function of temperature for SK/PRO/P407_3_ hydrogel.

**Figure 2 pharmaceutics-14-02039-f002:**
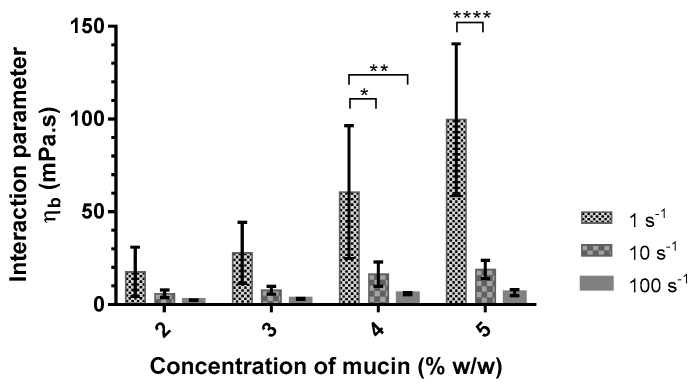
Interaction parameter values (η*_b_*) of alginate (Protanal LF 10/60 at 0.10% *w/w*) with different concentrations of mucin (2–5% *w/w*) in artificial saliva as a function of shear rate (1, 10, and 100 s^−1^) at 37 °C (mean ± SD, n = 3). *: *p* < 0.05; **: *p* < 0.01; ****: *p* < 0.0001.

**Figure 3 pharmaceutics-14-02039-f003:**
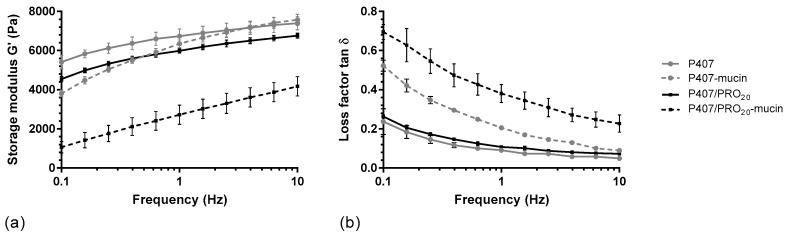
Rheological profiles of P407 and P407/PRO_20_ solution, in absence (P407, P407/PRO20) or presence of mucin (P407–mucin, P407/PRO20–mucin): (**a**) storage moduli (G′) and (**b**) loss factors (tan δ) as function of frequency (mean ± SD, n = 3). PRO: Protanal LF 10/60.

**Figure 4 pharmaceutics-14-02039-f004:**
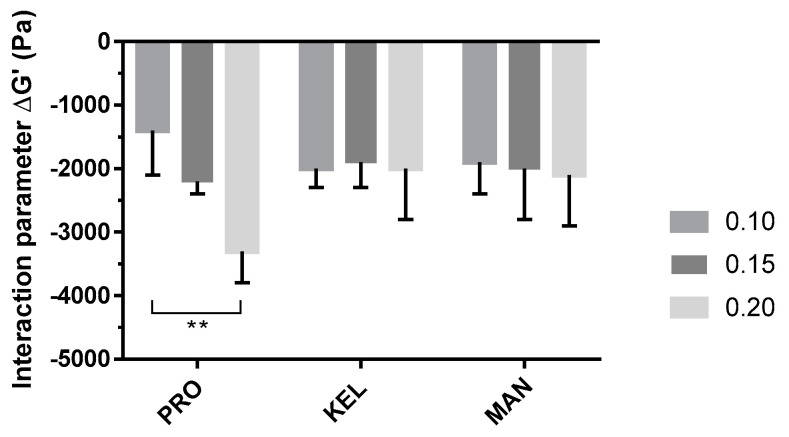
Impacts of the alginate grade and concentration (0.10, 0.15, and 0.20%, *w/w*) on interaction parameter (ΔG′) evaluated with mucin (5%, *w/w*) in artificial saliva. Data are reported as the mean ± SD (n = 3). PRO: Protanal LF 10/60; KEL: Keltone LVCR; MAN: Manucol DH. **: *p* < 0.01.

**Figure 5 pharmaceutics-14-02039-f005:**
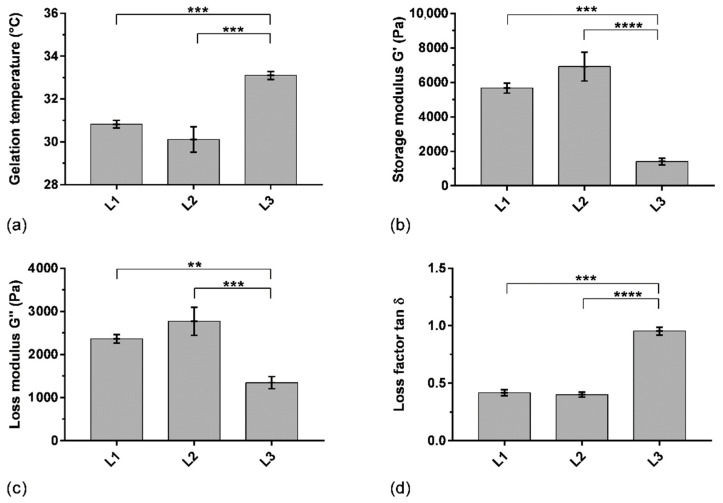
Rheological parameters of SK/PRO/P407 hydrogel using three batches of P407 (L1, L2, L3): (**a**) gelling temperature, (**b**) storage modulus at 37 °C, (**c**) loss modulus at 37 °C, (**d**) loss factor at 37 °C. Means and standard errors of three experiments are represented. **: *p* < 0.01; ***: *p* < 0.001; ****: *p* < 0.0001.

**Figure 6 pharmaceutics-14-02039-f006:**
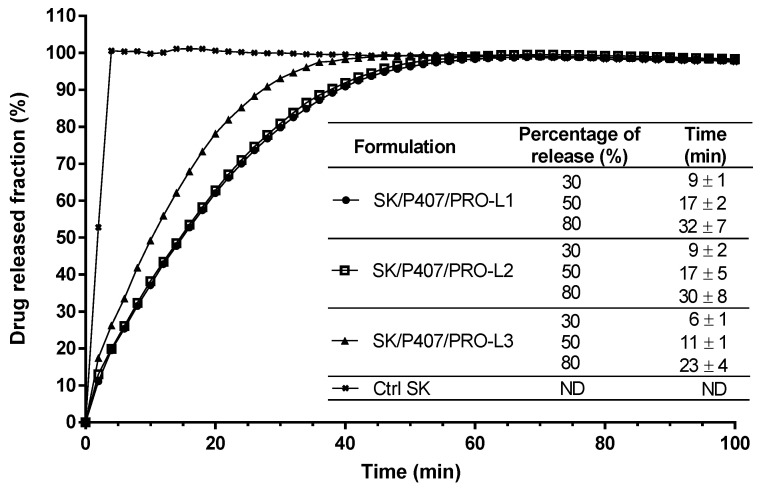
In vitro SK release profiles from SK solution in water (Ctrl SK) or SK/P407/PRO hydrogels formulated with three batches of P407 (L1, L2, or L3) using a flow-through USP-4 apparatus. The means and standard errors of three experiments are represented, and the times required to observe 30, 50, and 80% of drug release have been specified. (ND: not determined).

**Figure 7 pharmaceutics-14-02039-f007:**
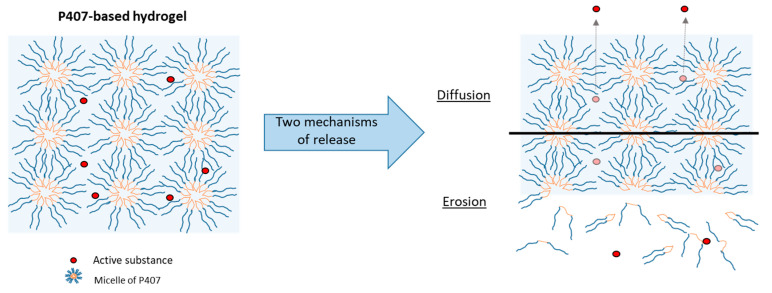
Schematic representation of two release mechanisms (diffusion and erosion) from a P407-based hydrogel.

**Figure 8 pharmaceutics-14-02039-f008:**
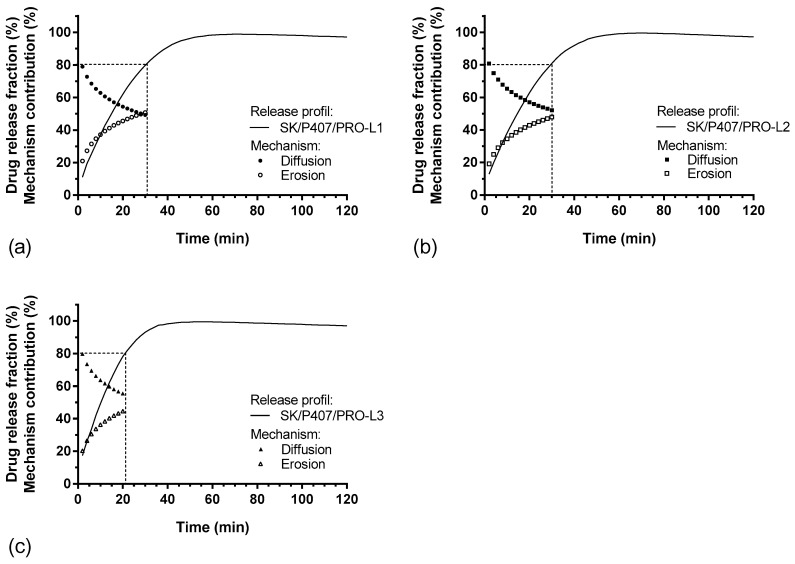
In vitro SK release profiles and percentages of diffusion and erosion contributions (%) as function of time: (**a**) SK/P407/PRO-L1, (**b**) SK/P407/PRO-L2, (**c**) SK/P407/PRO-L3.

**Table 1 pharmaceutics-14-02039-t001:** Composition of M and G residues for different grades of sodium alginate and range of viscosities.

Type of Alginate	Viscosity ^(1)^ (mPa·s)	M/G Ratio	Reference
Protanal LF 10/60	30–60	0.4–0.7	[17]
Manucol DH	40–90	1.5–1.8	[18,19]
Keltone LVCR	100–300 (2%)	1.5–2.3	[17]

^(1)^ 1% solution unless specified.

**Table 2 pharmaceutics-14-02039-t002:** Proportions of polymers (P407, alginates) and (*S*)-ketamine hydrochloride (SK) in hydrogels.

Sample Name	Batches of P407	Concentration of P407 (% w_P407_/w_water_)	Grades of Alginate	Concentration of Alginate (% *w/w*)	Concentration of SK (% *w/w*)	Use
SK/PRO/P407_1_	L1	15.0	PRO	0.10	9.23	Rheological study
SK/PRO/P407_2_		15.5				
SK/PRO/P407_3_		16.0				
SK/PRO/P407_4_		16.5				
SK/PRO/P407_5_		17.0				
P407	L1	16.0	-	-	-	Mucoadhesion
P407/PRO_10_	L1	16.0	PRO	0.10		
P407/PRO_15_				0.15		
P407/PRO_20_				0.20		
P407/KEL_10_	L1	16.0	KEL	0.10		
P407/KEL_15_				0.15		
P407/KEL_20_				0.20		
P407/MAN_10_	L1	16.0	MAN	0.10		
P407/MAN_15_				0.15		
P407/MAN_20_				0.20		
SK/PRO/P407-L1	L1	16.0	PRO	0.10	9.23	Rheological study and in vitro drug release
SK/PRO/P407-L2	L2					
SK/PRO/P407-L3	L3					

**Table 3 pharmaceutics-14-02039-t003:** Equations and parameters of mathematical models.

Models	Equations	Parameters	Numbering
Higuchi	$\frac{Mt}{M\infty }=k_{H}\times t^{1/2}$	$k_{H}$	(5)
Korsmeyer–Peppas	$\frac{Mt}{M\infty }=k_{KP}\times t^{n}$	$k_{KP},\;n$	(6)
Hopfenberg	$\frac{Mt}{M\infty }=100\times [1-\left(1-k_{HB}\times t{)}^{n}\right]$	$k_{HB},\;n$	(7)
Peppas–Sahlin	$\frac{Mt}{M\infty }=k_{1}\times t^{1/2}+k_{2}\times t^{2}\;$	$k_{1},\;k_{2}$	(8)
Makoid–Banakar	$\frac{Mt}{M\infty }=k_{MB}\times t^{n}\times e^{-kt}\;$	$k_{MB},\;n,\;k$	(9)

*Mt*/*M*∞: fraction of drug release at every time point *t*; *k_H_*: Higuchi release kinetic constant; *k_KP_*: kinetic constant; n: release exponent *k_HB_*: kinetic constant; *k*_1_: diffusion constant; *k*_2_: relaxation (erosion) constant; *k_MB_*: kinetic constant.

**Table 4 pharmaceutics-14-02039-t004:** Viscosities (mPa·s) of mucin solution (η*_m_*) and alginate–mucin mixture (η*_t_*) at a shear rate of 1 s^−1^ and the calculated interaction parameter (η*_b_*). For η*_b_* calculation, viscosities of alginate (η*_p_*) (Protanal LF 10/60 at 0.10%, *w/w*) in phosphate buffer or artificial saliva were η*_p_* = 1.5 ± 0.9 mPa·s and 1.9 ± 0.2 mPa·s at 1 s^−1^, respectively. Data were reported as the mean ± SD (n = 3). Significant differences between viscosities in artificial saliva and phosphate buffer and same concentration of mucin are highlighted with *: *p* < 0.05; ***: *p* < 0.001; ****: *p* < 0.0001.

Aqueous Media	Concentrations of Mucin (%*w/w*)	η*_m_* (mPa·s)	η*_t_* (mPa·s)	η*_b_* (mPa·s)
Phosphate buffer	2	3.5 ± 0.3	4.5 ± 0.8	0.7 ± 1.4
	3	6.0 ± 0.8	8.7 ± 1.4	1.2 ± 0.1
	4	9.3 ± 0.4	15.2 ± 1.4 *	4.4 ± 1.1 *
	5	14.7 ± 2.1	20.7 ± 4.7 ****	4.5 ± 1.8 ***
Artificialsaliva	2	7.3 ± 6.1	26.9 ± 10.5	17.6 ± 13.3
	3	8.3 ± 1.8	38.1 ± 18.2	27.8 ± 16.5
	4	12.1 ± 1.6	74.6 ± 35.6 *	60.6 ± 35.8 *
	5	19.3 ± 1.4	121.0 ± 41.0 ****	99.7 ± 40.9 ***

**Table 5 pharmaceutics-14-02039-t005:** Mathematical model fitting of drug release data from SK/P407/PRO formulated with three batches of P407 (L1, L2, or L3).

Mathematical Models	Parameters	Batches of P407		
		L1	L2	L3
Higuchi	$k_{H}$	13.556	13.774	16.281
	R^2^_adj_	0.941	0.947	0.950
	AIC	91.6	90.1	54.5
Korsmeyer–Peppas	$k_{KP}$	7.347	7.772	10.261
	n	0.707	0.693	0.680
	R^2^_adj_	0.999	0.999	0.999
	AIC	34.3	29.0	16.1
Hopfenberg	$k_{HB}$	0.008	0.008	0.005
	n	5.449	6.050	13.589
	R^2^_adj_	0.997	0.995	0.990
	AIC	49.4	56.9	39.1
Peppas–Sahlin	$k_{1}$	7.384	7.884	9.799
	$k_{2}$	1.385	1.322	1.767
	R^2^_adj_	0.997	0.998	0.999
	AIC	48.2	42.4	18.8
Makoid–Banakar	$k_{MB}$	6.193	6.963	8.638
	k	0.006	0.004	0.000
	n	0.814	0.763	0.825
	R^2^_adj_	1.000	0.999	0.998
	AIC	22.1	28.1	49.2

*k_H_*: Higuchi release kinetic constant; R^2^_adj_: Adjusted coefficient of determination; AIC: Akaike information criterion; *k_KP_*: Kinetic constant; *n*: Release exponent *k_HB_*: Kinetic constant; *k*_1_: Diffusion constant; *k*_2_: Relaxation (erosion) constant; *k_MB_*: Kinetic constant.

## Data Availability

Not applicable.

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
