# Peer review of "Development of Thermosensitive and Mucoadhesive Hydrogel for Buccal Delivery of (S)-Ketamine"

_pharmaceutics, 2022, doi:10.3390/pharmaceutics14102039_

Round 1
Reviewer 1 Report
This paper deals with the development of mucoadhesive hydrogels based on poloxamer 407 and sodium alginate for the local release of (S)-ketamine in the buccal cavity. Interesting results are presented, the following issues are requested to the authors.
1. Different grades of alginate and batch of poloxamer 407 were used in present study. The composition of residues for alginate samples was provided in Table 3. Please, also provide the principal differences between poloxamer 407 samples of different batch.
2. Figure 1, include information of solvent (buffer pH) used in gelling temperature study and the sol-gel temperature of neat poloxamer 407.
3. Modify the size of axis title and labels in Figure 3 and 6.
4. Figure 3, represent and discuss the relative behavior of storage/loss modulus as function of frequency.
5. Figure 5, the column bars are missing.
6. Drug carriers in gelling forms are usually intended to prolong the drug release at the administration site. In this work, the anesthetic was released from hydrogel formulation in the first 5 min. Please, include more arguments to support the practical application of the present drug delivery system.
Author Response
We thank the reviewer for these helpful comments. We hope this new version of the manuscript was improved by these changes. Please open the attachment to see the changes (highlighted green) in the manuscript.
- Different grades of alginate and batch of poloxamer 407 were used in present study. The composition of residues for alginate samples was provided in Table 3. Please, also provide the principal differences between poloxamer 407 samples of different batch.
Response 1: The properties of the alginates were specified because they are different products. The characteristics of the batches of P407 are claimed to be equivalent by the supplier. It is the same product and the same supplier. The supplier's specifications are therefore the same for the three batches. This is why we did not specify the differences between P407 samples. The underlined mentions “Three different batches of the same grade of P407 (L1, L2, L3) from the same supplier” were added in 2.2 (lines 112 to 113).
- Figure 1, include information of solvent (buffer pH) used in gelling temperature study and the sol-gel temperature of neat poloxamer 407.
Response 2: All components of SK/PRO/P4071-5 were dissolved in water (mention in 2.2, line 121). No other solvent or buffer were used for gelling temperature study. The mention “in water” in the title of Figure 1 could suggest a dilution of the sample whereas it is not the case. No mention was added in the title of Figure 1. The underlined mention “The gelling temperature (Tsol-gel) of the samples was determined by oscillatory measurements” was added in 2.4 (line 144).
- Modify the size of axis title and labels in Figure 3 and 6.
Response 3: All figures were revised to homogenize the format (size of axis title, labels, resolution).
- Figure 3, represent and discuss the relative behavior of storage/loss modulus as function of frequency.
Response 4: The relative behavior of storage/loss moduli (loss factor) was included in Figure 3 and the title of this figure was modified with the underlined mentions “Figure 3. Rheological profiles of P407 and P407/PRO20 solution, in absence (P407, P407/PRO20) or presence of mucin (P407-mucin, P407/PRO20-mucin): (a) Storage moduli (G’) and (b) Loss factor (tan δ) as function of frequency”. The underlined mention “Storage (G’) and loss (G’’) moduli and loss factor (tan δ = G’’/G’) were measured for all samples after an equilibration time of 60 s to ensure a temperature adaptation.” was added in 2.5.2 (line 182). The citation in the text has been added (lines 346 and 358). The evolution of this parameter as function of frequency is similar to storage modulus. The mention “The increase of loss factor (Figure 3b)” was added in the text (line 358).
- Figure 5, the column bars are missing.
Response 5: The column bars were added to homogenize all figures.
- Drug carriers in gelling forms are usually intended to prolong the drug release at the administration site. In this work, the anesthetic was released from hydrogel formulation in the first 5 min. Please, include more arguments to support the practical application of the present drug delivery system.
Response 6: The only condition when the anesthetic was released under 5 min is the “Ctrl SK”, which corresponds to an (S)-ketamine solution in water (i.e., no excipients). For more clarity, the underlined mention was added in the legend of Figure 6 “In vitro SK release profiles from SK solution in water (Ctrl SK) or SK/P407/PRO hydrogels formulated with three batches of P407 (…)”
Gelling form can be used to prolong the drug release or for local administration. Here, we have chosen this form to prolong the residence time of the product in contact to buccal mucosa to optimize the locally absorption of SK and propose other routes than IV or IM. This information (underlined) was added in 4.Conclusion (line 489) “A mucoadhesive agent was incorporated in the P407-based hydrogel to optimize the residence time of this formulation on the buccal mucosa and then increase the absorption of SK through mucosa” and line 505 “3) to avoid tissue infraction caused by IV or IM injections (…)”

Reviewer 2 Report
Comments to the Authors
Manuscript number Pharmaceutics-1899386 “Development of thermosensitive and mucoadhesive hydrogel for buccal delivery of (S)-ketamine” presents the development of a thermoresponsive hydrogel based on P407, for a new way to dose the (S)-ketamine appropriate for buccal administration. For improve the gelling temperature the (S)-ketamine was associate with P407 and alginate. This type of formulation stays in liquid form at room temperature permitting an easy way to administrate and in gel form to extend the residence time on contact with buccal mucosa. Including the alginate in the formulation increased the interaction with mucin and determined a better adhesion to mucosa.
The Protanal LF 10/60 one of the three grades of alginates presented a different behaviour on the interaction with mucin. This difference can be explained by the differences of the ratio of guluronate component in alginate structure. In vitro drug release studies showed that this variability has no significant impact on drug release profile according to the similarity factor ƒ2. Mathematical models were also applied to understand the mechanisms at the origin of drug release. Diffusion was the principal mechanism related to the (S)-ketamine properties. That new formulation of (S)-ketamine is a interesting alternative to take charge of the acute pain. The transmucosal route is promising: to make higher the bioavailability of a drug sensitive to first-pass metabolism, to increase the onset of action as compared to per os administration, and to prevent tissue infraction and so superimposed pain and distress.
Here are some questions and advices for the authors:
-Maybe in introduction it is better to say something about the latest findings in the mechanisms which are at the origin of drug release: diffusion and erosion.
-Figure 3, 6, the resolution should be improved, are not clear.
-The equations should be written using the same style.
-A scheme of the mechanism and method described in the paper I think are useful.
If the manuscript would have some minor revision before publishing, it will be interesting for the readers of the Pharmaceutics.
Author Response
We thank the reviewer for these helpful comments. We hope this new version of the manuscript was improved by these changes. Please open the attachment to see the changes (highlighted green) in the manuscript.
- Maybe in introduction it is better to say something about the latest findings in the mechanisms which are at the origin of drug release: diffusion and erosion.
Response: To introduce the mechanisms, which are at the origin of drug release, a sentence was added in 1.Introduction after the description of poloxamer (lines 71 to 73): “Gelling form can be used to prolong the drug release (decrease of diffusion rate or slowly erosion) or to enhance the absorption of active substance in the case of local administration”.
- Figure 3, 6, the resolution should be improved, are not clear.
Response: All figures were revised to homogenize the format (size of axis title, labels, resolution).
- The equations should be written using the same style.
Response: For using the same style for all equations, numbers of equation were added in the table 3, The equations 5 to 9 were presented in a table for more clarity.
-A scheme of the mechanism and method described in the paper I think are useful.
Response: A schematic representation of the two mechanisms of release was added in 3.4: “Figure 7. Schematic representation of two release mechanisms (diffusion and erosion) from a P407-based hydrogel”.

Reviewer 3 Report
The manuscript by Thouvenin et al. reports the fabrication of thermosensitive and mucoadhesive hydrogel for buccal delivery of an active drug for acute pain. The research is interesting but some issues should be addressed before the publication. Below I report some comments:
1. The introduction clearly reports the goal of the work but it lacks of a comparison with similar systems already explored.
2. Further information regarding the three grades of sodium alginate should be reported in M&M sections (molecular weight for example). Information reported in table 3 should be moved to M&M section.
3. Which was the idea behind the programming of experimental campaign? Why only determined combinations of concentration of polymer and grade of SA were chosen? For which reason the amount of SK was fixed?
4. Statistical differences can be highlighted in table 4 by applying ANOVA test to show if the values are statistically significant.
5. Authors should use the same formatting for the figures reported in the manuscript. Figure 3 must be completely revised.
6. Figure 6 needs to be revised in compliant with the formatting of other figures.
7. Release profiled should start from t=0 minutes.
8. What about the slight difference of L3 sample? Why does the release kinetic seem to be faster? Could it be related to the viscosity of the system or other factors can affect it? How can they affect the released % of SK?
9. Is the diffusion phenomenon the controlling one in release profiles? The authors reported contrasting statements in section 3.4
10. Conclusions can be simplified by removing redundant statements and by reporting numerical results.
Author Response
We thank the reviewer for these helpful comments. We hope this new version of the manuscript was improved by these changes. Please open the attachment to see the changes (highlighted green) in the manuscript.
- The introduction clearly reports the goal of the work but it lacks of a comparison with similar systems already explored.
Response 1: A sentence was added in introduction, lines 57 to 62:“For buccal delivery, mucoadhesive forms such as tablets, patches or films were developed to increase the bioavailability of the active substances by reducing salivary losses and by optimizing contact and thus absorption. The disadvantage of these galenic forms is the difficulty of adapting the dose administered to the patient's weight. Liquid forms are more adapted, especially in children”.
- Further information regarding the three grades of sodium alginate should be reported in M&M sections (molecular weight for example). Information reported in table 3 should be moved to M&M section.
Response 2: The Table 3 was reported in M&M section (2.2). The different grades of alginate were defined by the viscosity of alginate solution indicated in Table 3. The supplier did not give further pertinent information, such as molecular weight, which could be used to differentiate these grades.
- Which was the idea behind the programming of experimental campaign? Why only determined combinations of concentration of polymer and grade of SA were chosen? For which reason the amount of SK was fixed?
Response 3: The idea of this project was the development of P407-based hydrogel for buccal administration of (S)-ketamine for acute pain.
Compatibility between P407 and SA is known and already described in literature. A sentence was added to explain this choice in 3.2 (lines 272 to 274): “Sodium alginate exhibits interesting properties and compatibility with P407, as synthetized in the review written by E. Giuliano et al. [29]”. Despite a known compatibility, we were interested in the guluronate/mannuronate studying which can modulate alginate’s gelation/interaction properties. Thus, we focused on concentrations and grades of alginate to highlight the importance of the grade, which is not commonly described in literature.
The concentration of (S)-ketamine was established accordingly to the recommended doses for a use in pediatric population (0.5-1.0 mg/kg for parenteral routes). The concentration of (S)-ketamine was chosen on two “general” criterions: 1) the concentration in the solgel must be compatible with the dosage interval 0.5-1.0 mg/kg, 2) for a low volume of administration in order to cover the buccal mucosa with a thin film allowing the maximum of (S)-ketamine transfer from the gel to the mucosa (i.e., high drug loading). Thus, the (S)-ketamine concentration was preliminary set to the half of its water solubility (i.e., near 20g/100mL accordingly to literature) to limit the risk of precipitation while allowing a high drug loading compatible with the dosage interval. No precipitation was observed in the described formulations, and excipients concentrations were chosen accordingly to fulfill the specified properties of an optimal solgel formulation (i.e. Tsol-gel near 30°C and a fast release under 1 hour but non-instantaneous). Thereby, different (S)-ketamine concentrations were not studied, and the work focused on excipients concentrations and grades. A sentence was added to explain the choice of (S)-ketamine concentration (lines 110 to 112): “The concentration of SK was set at 9.23% w/w to obtain a dose of 13 mg (0.5 mg/kg) for an administrated volume of 140 µL corresponding to one spray”.
- Statistical differences can be highlighted in table 4 by applying ANOVA test to show if the values are statistically significant.
Response 4: ANOVA test was applied and the significate differences were indicated in table 4.
- Authors should use the same formatting for the figures reported in the manuscript. Figure 3 must be completely revised.
Response 5: All figures were revised to homogenize the format (size of axis title, labels, resolution).
- Figure 6 needs to be revised in compliant with the formatting of other figures.
Response 6: All figures were revised to homogenize the format (size of axis title, labels, resolution).
- Release profiled should start from t=0 minutes.
Response 7: In Figure 6, all release profiles were modified to start from t= 0 min.
- What about the slight difference of L3 sample? Why does the release kinetic seem to be faster? Could it be related to the viscosity of the system or other factors can affect it? How can they affect the released % of SK?
Response 8: It seems that there is a slight difference between L3 and the two other samples. Indeed, this could be related to viscoelastic properties presented in figure 5 (lines 473 to 480). But these differences did not impact SK release accordingly to FDA/EMA specification on fast release dosage forms. Thus, the slight difference showed not significant impact considering recommendations on drug dissolutions testing.
- Is the diffusion phenomenon the controlling one in release profiles? The authors reported contrasting statements in section 3.4
Response 9: Indeed, it was a mistake of language because the diffusion is the predominant phenomenon at the origin of SK release. The text has been modified accordingly:
- line 444: “predominantly” was replaced by “only”: “Thus, the release kinetic does not appear to be only related to the diffusion phenomenon”
- In 4.Conclusion, a sentence has added (line 497): “Two phenomena, diffusion and erosion, was occurring in the hydrogel” (sentence in 4.Conclusion).
- Conclusions can be simplified by removing redundant statements and by reporting numerical results.
Response 10: Conclusion was simplified by removing redundant statements and numerical results were reported. The strikethrough text was removed and the underlined mentions was added (lines 482 to 497): “A thermoresponsive hydrogel based on P407 was designed to develop a new dosage form of (S)-ketamine suitable for buccal administration. The association of (S)-ketamine (9.23% w/w), P407 (16% w/wwater) and alginate (0.1% w/w) was studied to ensure an optimal gelling temperature around 31°C. […] A mucoadhesive agent was incorporated in the P407-based hydrogel to optimize the residence time of this formulation on the buccal mucosa and then increase the absorption of SK through mucosa. […]. Among the three grades of alginates, Protanal LF 10/60 showed a different behaviour on the interaction with mucin. This difference could be explained by the differences of the high ratio of guluronate component in alginate structure. Then, the impact of P407 batches was also studied and a significant batch-to-batch variability of P407 on rheological properties was observed. However, in vitro drug release studies demonstrated that this variability has no significant impact on drug release profile according to the similarity factor (ƒ2 > 50) […].

Round 2
Reviewer 3 Report
The authors have satisfactorily improved the manuscript according to the suggested comments. I recommend now the publication without further modifications.